# Novel Ion Channel Targets and Drug Delivery Tools for Controlling Glioblastoma Cell Invasiveness

**DOI:** 10.3390/ijms222111909

**Published:** 2021-11-02

**Authors:** Alanah Varricchio, Sunita A. Ramesh, Andrea J. Yool

**Affiliations:** 1School of Biomedicine, University of Adelaide, Adelaide, SA 5005, Australia; alanah.varricchio@adelaide.edu.au; 2College of Science and Engineering, Flinders University, Bedford Park, SA 5042, Australia; sunita.ramesh@flinders.edu.au

**Keywords:** glioma, brain cancer, glioblastoma, aquaporin, membrane intrinsic protein, K_V_ channel, Na_V_ channel, Ca_V_ channel, migration, motility

## Abstract

Comprising more than half of all brain tumors, glioblastoma multiforme (GBM) is a leading cause of brain cancer-related deaths worldwide. A major clinical challenge is presented by the capacity of glioma cells to rapidly infiltrate healthy brain parenchyma, allowing the cancer to escape control by localized surgical resections and radiotherapies, and promoting recurrence in other brain regions. We propose that therapies which target cellular motility pathways could be used to slow tumor dispersal, providing a longer time window for administration of frontline treatments needed to directly eradicate the primary tumors. An array of signal transduction pathways are known to be involved in controlling cellular motility. Aquaporins (AQPs) and voltage-gated ion channels are prime candidates as pharmacological targets to restrain cell migration in glioblastoma. Published work has demonstrated AQPs 1, 4 and 9, as well as voltage-gated potassium, sodium and calcium channels, chloride channels, and acid-sensing ion channels are expressed in GBM and can influence processes of cell volume change, extracellular matrix degradation, cytoskeletal reorganization, lamellipodial and filopodial extension, and turnover of cell-cell adhesions and focal assembly sites. The current gap in knowledge is the identification of optimal combinations of targets, inhibitory agents, and drug delivery systems that will allow effective intervention with minimal side effects in the complex environment of the brain, without disrupting finely tuned activities of neuro-glial networks. Based on published literature, we propose that co-treatments using AQP inhibitors in addition to other therapies could increase effectiveness, overcoming some limitations inherent in current strategies that are focused on single mechanisms. An emerging interest in nanobodies as drug delivery systems could be instrumental for achieving the selective delivery of combinations of agents aimed at multiple key targets, which could enhance success in vivo.

## 1. Introduction

Glioblastoma multiforme (GBM) is a primary astrocytoma that represents more than 60% of all intracranial tumors. The rapid growth of glioblastoma tumors and the ensuing pressure on the brain can manifest symptoms including chronic headaches and seizures, and impair motor function and cognitive processes depending on the location of the tumor mass [1]. Classed as a Grade IV glioma and most commonly found in the frontal, temporal, parietal and occipital lobes, glioblastoma is a malignant and frequently occurring type of brain tumor that is among the deadliest of human cancers [2,3]. The median survival expectancy of glioblastoma patients is only 12 to 14 months after diagnosis. Patient outcomes have only improved marginally, despite decades of effort on developing more powerful and intricate treatments [4]. An abysmal survival rate in glioblastoma patients, despite intense chemo- and radiotherapy treatments, renders the discovery of novel therapeutic interventions of paramount importance. The essential shortcoming of current approaches is that the rapid infiltration of glioma cells into other brain areas, the dominant factor which impacts survival, has not yet been addressed.

Pharmacological modulators of classes of AQPs and ion channels have been successfully used to inhibit cellular migration, invasion, and metastasis in various types of carcinomas in vitro and in vivo but remain comparatively unexplored in models of glioblastoma cell motility. Invasion and migration assays in vitro and murine xenograft models in vivo have demonstrated reductions in migration, invasion, and angiogenesis in cancers treated with pharmacological modulators of AQPs and ion channels [5,6,7,8,9,10,11,12,13]. Interfering RNA-knockdown methods have correlated cellular motility with pathways downstream of AQPs [14,15,16]. Tumor cell migration, invasion and metastasis are impaired by pharmacological blockers of AQPs and ion channels, but remain comparatively unexplored in glioblastoma [17].

The heterogeneous nature of glioblastoma tumors and likely overlap between different membrane signaling pathways are obstacles that impede the discovery of a simple single intervention. In parallel with the development of effective delivery tools to direct agents to tumor sites, a compelling area of new research focus is the development of optimal combinations of pharmacological agents that additively or synergistically act to halt glioblastoma motility, with minimal cytotoxic side-effects. Tailored combinations of drugs could offer new promise for designing low dose therapies that effectively limit glioblastoma motility without disrupting normal neural networks. Novel combinations used as adjunct therapies could aid success by extending the durations of effective windows for the administration of first line clinical treatments. 

## 2. Glioblastoma—A Daunting Clinical Challenge

Current glioblastoma treatment strategies combine radiation and chemotherapy with surgical resection where possible. Improvement of symptoms and small extensions in patient longevity are valuable [3], but these methods fall short of the goals for restoring a meaningful duration and quality of life. Diffuse infiltration of glioma cells into healthy brain tissue is the essential problem that makes glioblastoma difficult to control [17]. Treatment success is further complicated by the heterogeneous (multiforme) nature of glioblastoma tumors. An array of deletions, amplifications, and point mutations add complexity to the genetic profile of subpopulations of cells contained within same glioblastoma tumors, in turn conferring variable levels of resistance to therapies, and shifting the relative balances of multiple signal transduction pathways, all of which determine properties of viability, proliferation and motility [18]. Intra-tumor heterogeneity means that current treatments often can eliminate only a fraction of the glioblastoma tumor mass, leaving resistant cells that are able to propagate, spread, and drive relapses.

## 3. Hallmarks of Glioblastoma Subtypes 

A systematic correlation of phenotypes with molecular markers has identified a panel of diagnostic indicators, supporting the classification of subtypes of glioblastoma as proneural, neural, classical, and mesenchymal [19]. These molecular subtypes and distinguishing genetic signatures are summarized in Table 1.

These subtype-associated molecular markers do not necessarily drive glioblastoma pathology directly, but, as tools for classification of glioblastoma subtypes, they have aided segregation of the patterns of genetic changes that underlie resistance to clinical treatments. Mutations of anabolic signaling proteins phosphoinositide 3-kinase (PI3K), epidermal growth factor receptor (EGFR), and isocitrate dehydrogenase types 1 and 2 (IDH1, IDH2) have been widely detected in glioblastoma patient biopsy samples [21,22,23]. Via mechanisms illustrated in Figure 1, dysregulation of various protein signaling mediators and enzymes in glioblastoma tumors is thought to promote growth, proliferation, metabolism and angiogenesis, and build resistance to treatment. Activating missense mutations in proteins involved as PI3K catalytic and regulatory subunits (PIK3CA and *PIK3R1*), occur frequently in GBM [21,23]. PIK3CA encodes the catalytically active protein p110α, and *PIK3R1* encodes regulatory protein p85α, which form the PI3K complex (Figure 1A) [21,23]. GBM mutations mainly cluster around three residues involved in interactions between p110α and p85α, sterically hindering normal inhibitory contacts, and driving constitutive PI3K activity with concomitant increases in proliferation, growth, metabolism and angiogenesis [24].

EGFR is a membrane-spanning tyrosine kinase receptor that binds an array of ligands in the epidermal growth factor family. The receptor dimerizes on activation to regulate downstream signaling cascades for cellular proliferation, differentiation, adhesion, and migration; aberrant constitutive activity has been linked to multiple types of human malignancies [25]. In the EGFR gene, missense mutations, amplification, and rearrangement are commonly detected in glioblastoma tumors. The most frequent variant in GBM is EGFRvIII, found in 30–50% of GBM tumors. Deletion mutations can cause constitutive activation of the receptor (Figure 1B) [25,26]. Missense mutations primarily occurring in the ligand-binding domain can increase receptor autophosphorylation and activate downstream signaling cascades in anabolic pathways [25]. Truncation mutations in the EGFRvIII extracellular ligand-binding domain can (independently of ligand binding) induce upregulation of the transcription factor nuclear factor kappa B (NF-κB) and the pro-angiogenic chemokine interleukin IL-8, resulting in enhanced tumorigenic activity [26].

Structural alteration of active sites in isocitrate dehydrogenases IDH1 and IDH2 affects histone methylation patterns. When the oncometabolite 2-hydroxyglutarate occupies the active sites of histone demethylases, in place of α-ketoglutarate, histone demethylation occurs (Figure 1C) [27,28]. Mutation of Arg 132 to His reduced the production of α-ketoglutarate, but augmented the ability of IDH1 to convert α-ketoglutarate to 2-hydroxyglutarate [27,28]. Mutation of Arg 172 to Lys in IDH2 disrupted α-ketoglutarate binding to the active site, and prevented Ten-eleven translocation (TET) enzyme from catalyzing the hydroxylation of DNA 5-methylcytosine (5 mC) to 5-hydroxymethylcytosine (5 hmC) [29]. As seen for 2-hydroxyglutarate, 5 hmC disrupts histone demethylation and causes genome-wide alterations in histone and DNA methylation patterns that can promote malignant progression of gliomas [28]

PI3K, EGFR, IDH1 and IDH2 mutations are examples of factors that alter the glioblastoma epigenome to confer increased growth and resistance to therapy. In preclinical GBM models, the EGFR inhibitor osimertinib demonstrated efficient block of kinase activity in EGFR mutant lines. By binding irreversibly to the intracellular kinase domain, osimertinib significantly reduced GBM cell proliferation, migration and invasion [30]. Since EGFR mutants in GBM maintain a wild-type intracellular kinase domain [30], osimertinib binding remains unaffected by the presence of mutations in the extracellular domain. Discovering the clinical potential of osimertinib as a GBM treatment highlights the value in screening for resistance-conferring mutations in proteins targeted for pharmacological intervention. Multi-array profiling of the mutational status of the broad set of the proteins that are used for GBM motility could be a logical starting point for customized treatments, as a novel approach that ultimately could be implemented into decision algorithms for selecting optimal therapies for individual patients with glioblastoma.

## 4. Current Strategies for Glioblastoma Treatment 

Standard treatment for glioblastoma patients includes surgical resection of solid tumor where possible, combined with radiation and adjuvant oral chemotherapy. The primary chemotherapeutic drug administered to glioblastoma patients, temozolomide, is a prodrug activated by the alkaline environment of brain tumors to release highly reactive methyldiazonium cations, which methylate the purine bases of DNA [34]. Temozolomide-induced cell death is caused by methylation of the *O6* guanine residue (*O6*-MeG). If methyl adducts are not removed by methylguanine-DNA methyltransferase (MGMT), mispairing with thymine occurs during replication, triggering thymine excision. In the continuing presence of *O6*-MeG, recurring DNA strand breaks drive futile cycles of thymine reinsertion and excision, causing G_2_/M cell cycle arrest and ensuing apoptosis [35]. Low levels of endogenous MGMT are essential for temozolomide efficacy; an increase in MGMT activity is one of the mechanisms that leads to temozolomide-resistant phenotypes, in which therapeutic effects of alkylating agents are ablated. MGMT levels are inversely correlated with survival time in an astrocytoma (predominantly GBM) patient population, with median overall survival times of 8 months for high and 29 months for patients with low MGMT levels [36]. Patients with tumors rich in MGMT urgently need alternative therapeutic strategies, which could involve co-administration of a MGMT inhibitor, or discovery of agents that induce methyl adducts not susceptible to removal by MGMT.

The alkylating agent carmustine used to treat intracranial tumors works by a mechanism analogous to that of temozolomide, causing formation of cytosine-guanine diadducts [37]. A DNA repair enzyme, O^6^-alkylguanine-DNA alkyltransferase (AGT), which normally prevents this crosslinking event has been identified as a principal source of carmustine resistance, supported by clinical trial data showing a direct correlation between carmustine resistance and AGT activity levels in brain tumors [36,37]. Depleting AGT by pre-administration of the substrate *O*^6^-benzylguanine has been adopted as one strategy for restoring carmustine sensitivity. Glioma tumor xenografts in vitro and in vivo show increased therapeutic benefits when *O*^6^-benzylguanine is administered prior to carmustine [38,39]. However, dampening DNA repair systems with *O*^6^-benzylguanine limits the carmustine dosages that can be given safely to GBM patients, in order to retain adequate levels of normal repair [40,41]. Alternatively, giving carmustine at lower doses and less frequently runs the risks of permitting carmustine resistance development and GBM disease progression.

Erlotinib, a first-generation reversible inhibitor of receptor tyrosine kinases, is used to manage non-small-cell lung cancer and advanced pancreatic cancer [42]. Erlotinib blocks ATP binding to a site in the EGFR tyrosine kinase domain, preventing receptor autophosphorylation and the initiation of downstream signaling cascades that lead to cell growth, differentiation and survival (Figure 1B) [43]. Clinical trials with glioblastoma patients showed the responsiveness to erlotinib correlated with levels of EGFR expression, irrespective of temozolomide co-treatment. A total of 40% of patients with amplified EGFR experienced a 50% reduction in tumor area in response to erlotinib over a 30-month period, whereas only 14% of patients with non-amplified EGFR levels showed a positive response [44]. While this appears encouraging, the potential clinical value of erlotinib appears to be limited to GBM tumor subtypes that have growth dependent on amplified EGFR.

When standard alkylating chemotherapeutics (described above) prove unsuccessful, glioblastoma can be treated with bevacizumab, a monoclonal antibody which binds circulating vascular endothelial growth factor (VEGF) to inhibit angiogenesis, known as anti-VEGF therapy [45,46]. Reduced microvascular growth limits blood supply to tumor tissues, increases vascular permeability and favors tumor endothelial cell apoptosis [47]. Alternative pathways in glioblastoma cells that promote angiogenesis constitute a mechanism of resistance to bevacizumab; for example, overexpression of the apoptosis inhibitor, survivin, in tumor endothelial cells reduces drug-induced apoptosis [48,49]. A substantial risk associated with antiangiogenic agents such as bevacizumab is hemorrhage, frequently reported for cases of central nervous system tumors [50]. Intracranial hemorrhages were reported in 32% of patients with recurrent glioblastomas, treated fortnightly with intravenous infusions of bevacizumab, as part of a Phase II study [51]. Bevacizumab can be beneficial in GBM patients for restraining angiogenesis and thus tumor growth, but the hemorrhaging risk associated with anti-angiogenic agents remains a major concern, and highlights the need for development of additional types of therapeutic interventions.

Innovative developments for glioblastoma treatment await tools for working around resistance-conferring factors such as enhanced MGMT, AGT, survivin levels and amplified EGFR activity. Personalized interventions for individual patients, designed based on custom profiling of the genetic and proteomic GBM signatures, could become a viable approach for slowing progression and overcoming resistance, but work will be needed to define all the key factors. Controlling glioblastoma invasiveness is a critical but underdeveloped component of available treatment strategies. The lack of success in identifying new drug targets over the last decade highlights the urgent need for new approaches and new alternative mechanisms. Glioblastoma treatments aimed at mechanisms beyond DNA repair are needed. Impeding glioblastoma tumor dispersal by targeting the membrane channels that function in key cellular motility pathways could constitute a fresh angle for development of adjunct therapies, which could be applied in parallel with existing procedures to buy time for standard therapy administration.

## 5. Cellular Motility as a Therapeutic Target in GBM

Glioblastomas rarely metastasize beyond the central nervous system, bounded by physical limits of the blood–brain barrier [52]; nevertheless, the exceptionally invasive nature of these tumors within brain parenchyma constitutes a major challenge for effective treatment. Cancer metastases cause of 90% of cancer-related deaths [53], but methods for management of aberrant cell motility remain scarce. Main-line drugs, illustrated by temozolomide, carmustine, bevacizumab and erlotinib, have been focused on inhibiting angiogenesis and proliferation, and inducing apoptosis. Pharmacological targeting of proteins involved in rate-limiting steps of cellular motility showed promise in in vitro models and preclinical trials, but substantial research in this area is still needed.

Invasion, defined as the direct penetration of motile cancer cells through extracellular matrix (ECM) into neighboring tissues [54], is distinct from metastasis, in which cells dissociate from a primary tumor and establish secondary tumors in distant locations [55]. As summarized in Figure 2, invasion involves a series of steps: cell polarization, protrusion, cell-matrix adhesion, ECM degradation and trailing edge retraction [56]. Main routes of invasion for glioblastoma cells through the human brain are myelinated tracts, basement membranes of blood vessels, pial surfaces, and subependymal layers surrounding ventricles [57]. Phenotypic expansion of GBM high-grade tumor capabilities have been associated with novel mutations in key genes, including for example the acquisition of neuron-like signaling processes that facilitate cell survival and invasion in gliomas with IDH and histone H3 mutations [58,59]. Microglia and macrophages associated with gliomas also display altered genetic profiles, with malignant phenotypes that enhance glioma invasiveness and resistance to chemo- and radiotherapies [60,61,62,63].

Integrins, a class of membrane proteins mediating cell-cell and cell-substrate adhesions, have been of interest as candidate targets for inhibition of metastasis in a variety of cancers. A cyclic peptide cilengitide, developed in the early 1990s, was of particular interest for glioblastoma and advanced through full clinical trials, but ultimately was not found to have consistent effects on survival outcomes [64]. Cilengitide mimics an arginine-glycine-aspartic acid motif recognized by integrins such as αvß3 and αvß5, acting at nanomolar or lower concentrations to block the integrin binding site and impair endothelial cell migration and angiogenesis [65]. Early work that prompted initial interest showed patients with recurrent glioblastoma receiving cilengitide twice weekly had a 6-month progression-free survival rate of 15% and a median overall survival of 9.9 months [66]. Cilengitide in combination with temozolomide and radiotherapy increased progression-free survival at 6 months by approximately 15% compared to standard therapy alone, but benefits appeared to be limited to patients with MGMT promoter methylation [67], a surprising finding, since MGMT status was not considered relevant to the biological activity of cilengitide [68]. Subsequent work segregated subjects by MGMT promoter methylation status, with positive carriers assigned to the Phase III CENTRIC trial, and negative to the Phase II CORE trial, but unfortunately outcomes showed no significant additional survival benefits in either [69,70]. Following assessment in approximately 30 different clinical trials for cancer, lack of efficacy against glioblastomas in Phase III trials terminated the arduous journey of cilengitide as a treatment for glioblastoma.

In summary, strategies for impeding glioblastoma progression that showed promise in early trials have benefits that appear to be limited to certain populations, or compromised by variability. Bevacizumab is effective in tumors rich in VEGF, erlotinib acts on tumors with high levels of EGFR, and. the anti-neoplastic effects of temozolomide are negligible in tumors with high levels of MGMT. Tailoring clinical interventions to glioblastoma subtypes might be necessary, and new combinations of treatments might prove more effective than single agents for the additional goal of reducing invasion. Knowing the key mechanisms involved in molecularly diverse subtypes of glioblastomas will be necessary to identify portfolios of candidate targets for therapies.

## 6. Aquaporins as Drug Targets in Glioblastoma Multiforme Tumors

Aquaporins (AQPs) are integral membrane proteins that facilitate fluxes of water and other solutes across biological membranes. Within the large major intrinsic protein (MIP) family, 15 mammalian AQP genes have been identified, with AQPs 0 to 12 expressed in higher mammals, and AQPs 13 and 14 found in older mammalian lineages [82]. Classical AQPs (0, 1, 2, 4, 5, 6 and 8) mediate fluxes of water, as well as gases, urea, ammonia, hydrogen peroxide, or charged particles depending on the subtype [83,84,85,86,87,88,89]. AQP1 exhibits cyclic-nucleotide-gated permeability to monovalent cations through the central tetrameric pore [89,90]. The aquaglyceroporins, AQPs 3, 7, 9 and 10, are permeable to glycerol and water, and in some subtypes mediate transport of urea, lactate or hydrogen peroxide [84,91,92,93,94,95,96]. Distantly related paralogs with only 20% homology to other mammalian subtypes are AQPs 11 and 12. AQP11 exhibits water, glycerol and H_2_O_2_ permeability [97,98,99,100]; the function of AQP12 remains to be determined.

AQP channels are organized as tetramers (Figure 3). Each monomer (~30 kD) has intracellular amino and carboxy termini, and six transmembrane α-helices (M1 to M6) connected by extracellular loops A and C, cytoplasmic loop D, and two hairpin domain loops B and E that fold together to create a water pore in each subunit. Loops B and E contain hallmark Asn-Pro-Ala (NPA) motifs, conserved across MIPs (Figure 3 inset 1). Asparagine side chains in NPA motifs act as hydrogen-bond donors and acceptors to coordinate single-file transport of water in classical AQPs, and water and glycerol transport in aquaglyceroporins [101]. A narrow region in the extracellular vestibule of the intrasubunit pore is framed by aromatic and arginine residues (termed the ar/R constriction; Figure 3 inset 2), which facilitates substrate selectivity by setting the limiting pore diameter [102,103].

In cancers, different classes of AQPs enhance motility by mechanisms (yet to be fully defined) that are associated with cell volume regulation, cell-cell and cell-matrix adhesions, interactions with actin cytoskeleton, control of proteases and extracellular-matrix degrading molecules, and colocalization with ion channels and transporters [71,72,73,74,75,76,77,79,80,81,104,105,106]. Quantitative analyses of transcriptomic data available at the Human Protein Atlas (https://www.proteinatlas.org accessed on 26 September 2021) showed transcript levels of certain classes of AQPs and ion channels in human glioblastoma biopsy samples (*n* = 153) are enriched (Figure 4), a classification defined as transcript levels at least four-fold greater than those in other tissues, measured as fragments per kilobase of transcript per million fragments mapped [24]. Enriched expression of AQPs 1, 4 and 9 in GBM fits with their proposed roles in glioma cell motility, proliferation, and survival [104,105,107,108,109,110]. 

AQP1, in particular, has been linked to rapid invasion and migration characteristic of aggressive cancers such as glioblastoma [111]. AQP1, localized at leading edges of GBM cells, is proposed to mediate water fluxes allowing shrinkage and morphological changes to facilitate invasion through the narrow extracellular spaces [106,112]. AQP1 facilitates glioma cell migration [112]. A unique property of AQP1, the gated cation channel activity, has been proposed to play a role in cellular migration. Pharmacological inhibitors of AQPs have been used to inhibit migration and invasion in cancer studies in vitro and in vivo. Bumetanide derivatives AqB007 and AqB011, which block the AQP1 nonselective cation channel in the central pore, reduced migration rates in colon cancer cell lines [11]. Bacopasides I and II from the medicinal plant *Bacopa monnieri* blocked colon cancer cell migration by targeting AQP1 [13]. Co-inhibition of both the water and ion pores in AQP1 achieved a greater loss of cellular motility than blocking water flux alone. Roles of the AQP1 dual water-and-ion channels in glioblastoma remain to be explored. AQP1 expression also influences matrix metalloproteases MMPs 2 and 9, connexin-43 and β-catenin to regulate adhesion and actin reorganization for membrane protrusion formation [104,110]. 

AQP4 channels are highly expressed in GBM, and have also been implicated in cell motility and migration [113,114]. AQP4 channel function is modulated by glutamate-mediated neural mechanisms of signaling and excitotoxicity [115]. Subcellular localization and the rates of protein turnover also govern AQP function [116,117], downstream of signaling pathways including vascular endothelial growth factor VEGF [118]. Lower levels of a small non-coding RNA miRNA-320a in gliomas were correlated with poor prognoses; conversely, miRNA-320a overexpression in U87MG and U251MG cell lines was associated with decreased levels of AQP4 and decreased migration and invasion [119]. Knockdown of AQP4 expression with siRNA resulted in impairment of glioblastoma cell migration and invasion in cell line LN229 [120]. Downregulated AQP4 has been linked with increased expression of β-catenin and connexin 43, reduced expression of matrix metalloprotease-2 (MMP-2), and decreased invasion [120]. The protein kinase-C (PKC) activator phorbol 12-myristate 13-acetate (PMA) suppresses migration and adhesion in glioma cells in vitro [104], and reduces AQP4–mediated water permeability. PKC at the leading edges of motile tumor cells regulates integrin trafficking, MMP expression and secretion, further enhancing tumor cell locomotion [93,105,108,112]. PMA decreased cell invasion in vitro, and when intracranially injected into immuno-deficient mice [105]. AQP4-mediated volume changes needed for cell growth and division could be regulated by PKC. In sum, the expression and localization of AQP4 at the leading edges of migrating glioma cells support the idea that AQP4, similar to AQP1, could be a target of interest for novel drug-based GBM treatments.

Permeability of AQP9 to lactate, glycerol, urea and hydrogen peroxide [121,122] could facilitate glioblastoma survival by counteracting lactic acidosis that accompanies hypoxic conditions within glioblastoma tumors [123,124]. Conversely, a tumor suppressing activity of AQP9 has been suggested in hepatocellular carcinoma, associated with inhibition of the Wnt/β-catenin pathway that reduces migration and proliferation [125,126]. Dysregulation of Wnt/β-catenin signaling appears to be unique to the proneural and mesenchymal subtypes of glioblastoma [127]. AQP9 has been implicated in both activation and inhibition of cellular motility, depending upon the anabolic signaling pathways in operation. Reduced AQP9 expression can compromise protrusion formation; AQP9 transfection in HEK-293 cells enabled formation of bleb-like membrane protrusions, which were blocked by the AQP9 antagonist HTS13286 [128,129]. Conversely, overexpression of AQP9 reduced migration and invasion in hepatocyte carcinoma cells, an effect that was reversed by Wnt/β-catenin activation [125]. GBM responses to the AQP9 inhibitor would be interesting to test in classical and neural glioblastoma subtypes, which are not appreciably affected by dysregulation of Wnt/β-catenin. If increased expression of specific AQPs and ion channels could be identified as a ubiquitous feature in GBM across the multiforme diversity of tumor subtypes, controlling malignant cell phenotypes by exploiting their dependence on anabolic metabolism and high motility could make these channels important co-targets for pharmacological inhibition in combined treatments.

## 7. Potential for Combined Inhibition of AQPs and Ion Channels

Ion channels are instrumental for signal transduction and maintenance of physiological homeostasis. As summarized in Table 2, various classes of ion channels also have been implicated in pathological changes in motility that accelerate invasion, migration and metastasis in malignant tumors.

Mean transcript levels of subtypes of acid-sensing, and voltage-gated potassium, sodium, and calcium channels (Figure 4) are amplified in glioblastoma patient biopsy tissues [24]. Pharmacological inhibitors of voltage-gated potassium (K_V_), sodium (Na_V_), and calcium (Ca_V_) channels, as well as chloride and acid-sensing ion channels, have been reported to impair cell survival, migration, and invasion in vitro in glioblastoma models.

A link between motility and K_V_ channel function was suggested by effects of the K_V_ channel blocker 4-aminopyridine, which correlated with downregulation of microRNA miR-10b-5p [136]. Overexpression of a miR-10b-5p mimic in glioma cells increased motility by upregulating RhoC (a regulator of focal adhesion assembly) and MMP2; all these effects were reversed by an inhibitor of miR-10b-5p [130]. Cellular motility was also affected by pharmacological inhibition of Na_V_ and Ca_V_ channels in glioblastoma models. The Na_V_ channel blocker tetrodotoxin inhibited invasiveness of prostate and breast cancer cell lines by 30 to 50% [5,6,20] and increased adhesion in highly metastatic prostate cancer cells [137]. Calcium channel inhibitors diltiazem and verapamil reduced squamous cell carcinoma invasion in vitro by reducing calcium influx and hindering EGFR-dependent invasion pathways; fendiline which inhibits both calcium channels and calmodulin decreased migration in pancreatic cancer cell lines by disrupting cell–cell adhesion mediated by β-catenin [14,135].

Peptide compounds isolated from arthropod venoms such as spider and scorpion toxins have been explored as potent inhibitors of cell locomotion. Na_V_ channels, upregulated in highly invasive malignant cells, were inhibited by an analgesic-antitumor peptide blocker (AGAP), a toxic polypeptide from scorpion which acts selectively on a subtype of sodium channels containing the β1 subunit [22] that is expressed in human glioma biopsies [24]. The scorpion toxin KAaH1 reduced cell migration by 27% in primary glioblastoma cell lines in vitro, via blockade of K_V_ channels. When combined with a related scorpion peptide KAaH2, migration was inhibited by 60%, despite the absence of anti-migratory effects of KAaH2 when administered alone [138]. The scorpion toxin chlorotoxin triggered internalization of a cell-surface complex containing MMP2 and the colocalized chloride channel ClC-3, also disrupting glioma cell volume regulation [131], and reduced invasiveness by 34% in acutely dissociated glioma cells from human biopsy tissue, and by 55% in cultured glioma cell lines [106,139]. Chlorotoxin preferentially binds to neuroectodermal tumors such as glioblastoma, with no effect on invasion in human melanoma, astrocyte or fibroblast cells in vitro [139], a property that is of interest for the development of glioblastoma therapeutics. Radio-iodinated TM-601, a synthetic form of chlorotoxin, reduced tumor growth with minimal toxic side effects in Phase I clinical trials [140]. Ongoing Phase I/ll trials focus on improving TM-601 efficacy through conjugation of TM-601 to apoptosis-triggering molecules. Participant recruitment for a Phase I trial that will investigate chimeric antigen receptor T-cells carrying a chlorotoxin targeting domain as a treatment for recurrent and progressive glioblastoma tumors is currently underway (NIH ClinicalTrials.gov Identifier NCT04214392; information is available at https://clinicaltrials.gov/ct2/show/NCT04214392 accessed on 29 October 2021). Selective inhibition of acid-sensing ion channels with spider toxin psalmotoxin-1 attenuated invasion in glioma cell lines [141,142], and disrupted volume recovery responses in glioma cells after shrinkage in hyperosmolar solutions [132]. Recombinant peptides used as ion channel-specific blockers could offer an invaluable avenue to achieve target selectivity for new treatments to control cell migration and invasion.

As summarized in Table 2, pharmacological inhibition of Na_V_ and Ca_V_ function also caused apoptosis and reduced proliferation in glioblastoma. Calcium and sodium signals are important for cell cycle regulation and proliferation. The Ca_V_ channel inhibitor mibefradil inhibited primary glioblastoma stem cell proliferation and induced cell death [143], perhaps in part by indirect effects on growth factor and survivin levels [143,144]. In glioblastoma cell lines, block of sodium channels with cardiac glycosides digoxin and ouabain yielded an approximate 2-fold inhibition of proliferation compared to untreated controls, accompanied by detachment and apoptosis within 24 h [145]. The efficacy of clinical interventions that target oncogenic signaling pathways is often limited by eventual resistance to inhibitor therapy. Therapeutic effects of current major glioblastoma treatments are curtailed in tumors overexpressing DNA damage repair systems or operating alternative angiogenic or apoptotic control pathways [36,37,48,49]. Cancer cells exploit signaling pathway redundancy to escape disruption by therapeutic agents. Combinations of agents could counteract the advantage created by overlapping signaling mechanisms.

Promising data for improved efficacy of combined inhibitors are emerging for aquaporins. Bacopasides I and II isolated from the medicinal plant *Bacopa* have been identified as blockers of AQP1 channels, and shown to slow colon cancer cell migration [13]. The finding that bacopasides block AQP1 has been successfully translated into a novel treatment in vivo, with recent work showing that cardiac hypertrophy induced by AQP1-mediated fluxes of hydrogen peroxide H_2_O_2_ was prevented by systemic treatment with bacopaside II [146]. In breast cancer cells, a combination of both bacopasides, I which blocks the AQP1 ion channel, and II which blocks the AQP1 water pore, decreased invasion by up to 97%, an effect that was dependent on AQP1 expression [12]. Block of the AQP1 ion channel by the antagonist AqB011 reduced colon cancer cell migration up to 50% [11], but combined treatment with bacopaside II boosted the inhibitory effect to 81% block of migration and impaired lamellipodial formation, effects not observed when AQP1 membrane protein levels were low [10]. These results support the proposed value of restricting both water and ion transport for intervention in tumor cell motility, but further experiments in glioblastoma are needed to comprehensively survey the effects of blocking multiple water and ion channel types. Blocking AQPs has been shown to be effective in slowing cancer cell invasion, but effects of the combined inhibition of ion channels and AQP water and solute channels remain to be tested.

## 8. Use of Nanobodies to Overcome Off-Target Effects in the Brain Environment

Off-target effects arising from the extensive distribution of candidate protein targets such as AQPs and ion channels throughout the central nervous system present a major obstacle in terms of efficacy and safety of new pharmacological treatments. This limitation could be addressed in part by efficiently delivering pharmacological agents directly to tumor cells in vivo, rather than via systemic application. Antibodies are natural examples of biological targeting agents with high specificity (Figure 5).

Monoclonal antibodies (mAbs) such as bevacizumab have revolutionized the treatment of solid and hematological malignancies. The mAbs consist of two heavy and two light chains, each non-covalently associated with variable domains containing unique antigen binding sites. These paratopes confer target specificity and endow antibody diversity that enables the immune system to protect against a virtually unlimited array of antigens [147]. Rituximab, a monoclonal antibody used for patients with CD20-positive non-Hodgkin’s lymphoma or chronic lymphocytic leukemia, tags CD20 on the surface of leukemia and lymphoma cells, marking them for immune-mediated destruction without impacting stem cell progenitors needed for regeneration after treatment [148,149].

Some antibody–drug conjugates and unconjugated antibodies have received US Food & Drug Administration approval for delivering drugs to cancer cells. Taking advantage of the high rates of macromolecule recycling that enable the high proliferation rates seen in cancer cells [150,151], drugs can be designed to be activated by the abundant lysosomal proteases responsible for protein hydrolysis [152]. For example, upon exposure to proteolytic enzymes, the antibody–drug conjugate brentuximab–vedotin releases auristatin E, an apoptosis-inducing monomethyl that causes tumor regression in 86% of CD30-positive Hodgkin’s lymphoma and anaplastic large-cell lymphoma patients [153].

Despite successes in reducing angiogenesis and controlling white blood cell cancers, the broader applicability of mAbs and their drug conjugates is limited. With four polypeptide chains, antibody masses exceed 150 kD, hindering access to tumor cells, particularly solid tumors [147]. Diffusion rate is inversely proportional to the cube root of molecular weight. Poor diffusion of mAbs into tissues constrains their therapeutic potential, particularly in solid malignancies. Alternatives with low molecular weights, in vivo stability, clinically useful pharmacokinetic profiles, and high binding specificities characteristic of monoclonal antibodies are of keen interest.

A new generation of tumor therapeutics is arising from the serendipitous discovery of heavy chain antibodies (hcAbs; Figure 5), naturally produced by all members of the camelid family. The hcABs exhibit high target specificity and binding capacities akin to those of mAbs, despite the absence of light chains and CH1 domains of heavy chains [154]. The two heavy chains constituting hcAbs are each associated with antigen-binding domains termed nanobodies (Nbs; Figure 5). These nanobodies contain the complementarity-determining regions that contribute to target diversity and specificity [147]. The strongly hydrophilic structure of nanobodies confers superior stability and solubility in vivo and overcomes aggregation problems associated with mAbs and mAb–drug conjugates [147]. Nanobody biodistribution after in vivo administration can be largely restricted to the tumor site, with small molecular size facilitating rapid extravasation and penetration into tumor tissue, thus providing substantial advantages over their larger mAb counterparts. Poor diffusion of mAbs into tumor tissues is associated with wider dispersal and increased risks of off-target effects [155,156,157,158]. Nanobodies as drug antagonists and vehicles for drug delivery in solid tumor models, including glioblastoma, are an exciting area of new investigation.

Nanobodies used to target proteins and growth factors enriched in cancerous tumors have been shown to inhibit cancer growth and motility in vivo and in vitro. One example of a successful treatment involved inhibiting hepatocyte growth factor (HGF), which is overexpressed in most solid tumors and associated with tumor aggressiveness and poor prognoses [159,160]. HGF-induced proliferation, invasion and metastatic responses result from activation of receptor tyrosine kinase c-Met and downstream signaling cascades [161,162]. Glioblastoma xenografts in nude mice showed up to 85% less tumor growth after approximately 10 weeks of treatment with anti-HGF nanobody 1E2-Alb8, and no growth with nanobody 6E10-Alb8, as compared to untreated mice [158]. Anti-EGF nanobodies have shown similar success in human squamous cell carcinoma. Acting by inhibiting EGF-induced receptor autophosphorylation and associated downstream anabolic pathways, anti-EGF nanobodies Ia1, L2–3.40 and IIIa3 completely blocked EGF-induced proliferation of squamous cell carcinoma cells in vitro. Nude mice hosting subcutaneous xenograft tumors of squamous carcinoma cells showed decreased tumor volumes after injections of nanobodies Ia1, L2–3.40 or IIIa3 as compared with placebo control after four weeks [156]. Nanobody Nb206 raised against mitochondrial translation elongation factor induced pronounced and specific cytotoxicity in glioblastoma stem cells and an established glioblastoma cell line, with no observed effects on astrocytes or neural stem cells [163]. Evidence thus far shows nanobody approaches are capable of inhibiting tumor growth in vivo. Specialized nanobody systems that target key components of glioblastoma motility pathways could prove invaluable for halting glioblastoma tumor progression in a clinical setting.

Nanobodies also show promise as in vivo tools for directing the delivery of cytolytic agents in the form of antibody-toxin conjugates, aimed at specific tumor markers such as carcinoembryonic antigen (CEA) and EGFR. For example, nanobody cAb-CEA5 displays high affinity for CEA, which is abundant in colorectal cancers [164]. cAb-CEA5 fused with an enzyme β-lactamase was localized selectively to the colorectal cancer xenografts in nude mice, then allowing systemically administered prodrug 7-(4-carboxybutanamido) cephalosporin mustard to be converted locally to cytotoxic phenylenediamine mustard at the tumor site [155,165]. Partial or complete tumor regression was achieved without the toxicity observed in mice treated with phenylenediamine mustard directly [155]. A nanobody targeted to EGFR and conjugated with the apoptosis-inducing ligand TRAIL (ENb2-TRAIL) has been explored in glioblastoma. Neural stem cells transduced with ENb2-TRAIL became capable of secreting TRAIL. When these neural stem cells were cocultured with glioblastoma cells, secreted TRAIL reduced EGFR activation in the glioblastoma cells. This reduction was observed in glioblastoma cells overexpressing wild-type EGFR or expressing mutant EGFRvIII [157]. After simultaneously implanting EGFRvIII–expressing glioblastoma cells and TRAIL-secreting neural stem cells into nude mice, TRAIL-mediated apoptosis of glioblastoma cells ensued, and tumor invasiveness was suppressed by up to 60% [157]. This provides a promising means of overcoming the ligand-independent constitutive activity of EGFRvIII (see Figure 1B). EGFR-targeted nanobodies conjugated with inhibitors of invasion and migration in glioblastoma could lead new treatment strategies. Designing nanobody-based transport systems that deliver ion and water channel inhibitors as prodrugs to glioblastoma tumors are of interest and warrant preclinical investigation.

The blood–brain barrier poses a major challenge for the delivery of nanobody-conjugates to glioblastoma tumors. Efforts have been made to modify nanobodies to bind to receptors located on the endothelial cells comprising the blood–brain barrier, to trigger receptor-mediated transcytosis. Transferrin and insulin receptors have been of particular interest as tools to facilitate the transcytosis of molecular ‘Trojan horses’, in other words, therapeutic drugs fused to nanobodies against transferrin or insulin receptor, that can move the complex through the blood–brain barrier [166,167]. Targeting nanobodies to receptors that trigger transcytosis across the blood–brain barrier could be a promising method for the delivery of small molecule inhibitors into glioblastoma tumors.

Towards achieving this goal, the first line of investigation will require identification of the combinations of pharmacological agents for AQPs and ion channels that most potently suppress glioblastoma cell motility. Subsequent work would be directed toward designing nanobodies that directly and specifically inhibit AQP and ion channel activity in glioblastoma cells. Tumor-selective nanobodies would be expected to minimize the off-target effects that otherwise arise from indiscriminate systemic block of signaling and transport proteins expressed throughout the nervous system. An increasing library of glioblastoma biomarkers that distinguish brain tissue from glioblastoma cells will be essential [168,169]. Nanobody conjugates could carry inhibitors directly to glioblastoma tumor sites, improving specificity and limiting adverse side effects. Further confirmation of high target selectivity and absence of toxicity for nanobodies will be needed to establish their role in treating glioblastoma without compromising neuron and glial function, justifying their intriguing potential as novel therapeutics.

## 9. Conclusions

Glioblastoma multiforme has for decades remained one of the most difficult diseases to treat, resulting in poor patient prognoses despite administration of the best available therapies. Chemotherapy, radiotherapy and surgical resection reduce symptoms for patients and slightly extend longevity, but dispersal of glioma cells throughout brain parenchyma allows the tumor to escape control by therapies aimed mainly at eliminating primary tumors. Heterogeneity in glioblastoma tumors is an added challenge, in that populations of cells with resistance to alkylating agents and tyrosine kinase inhibitors remain viable and able to initiate relapses. The overview here of published work suggests that tools for impeding glioblastoma tumor invasiveness by concurrent targeting of key AQPs and ion channels could constitute a powerful adjunct in current GBM therapy. AQPs 1, 4 and 9, voltage-gated potassium, sodium and calcium channels, and acid-sensing ion channels are among candidate targets for reducing cancer cell motility. Combinations of agents conjugated to nanobodies which recognize glioblastoma biomarkers could provide an effective means of overcoming the difficulties created by overlapping functional roles of multiple signaling pathways, and minimizing toxic side effects on surrounding brain tissue. In response to the call for new therapeutic interventions for glioblastoma, we propose simultaneous GBM-targeted inhibition of water and ion movement could constrain tumor invasiveness, and augment the cancer therapy repertoire by extending the time of effective treatment for first-line therapies.

## Figures and Tables

**Figure 1 ijms-22-11909-f001:**
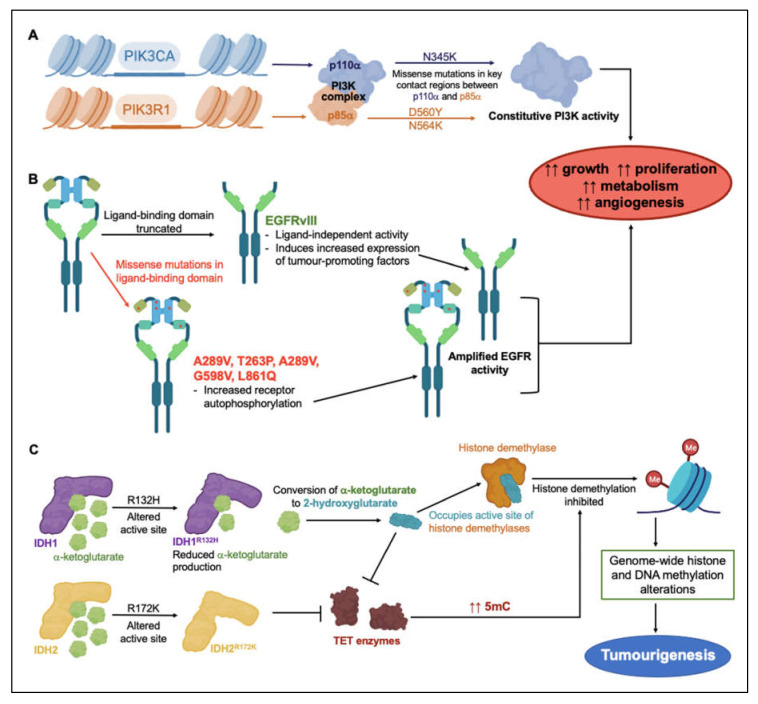
Overview of genetic events linked to treatment resistance strategies in glioblastoma malignancies. Mutagenetic events inducing constitutive (**A**) PI3K and (**B**) EGFR activity lead to increased tumor growth, proliferation, metabolism and angiogenesis [21,23,24,31,32,33]. (**C**) Mutations in substrate-binding active sites of IDH1 and IDH2 impede histone demethylation pathways, altering gene regulation and contributing to tumorigenesis [27,28].

**Figure 2 ijms-22-11909-f002:**
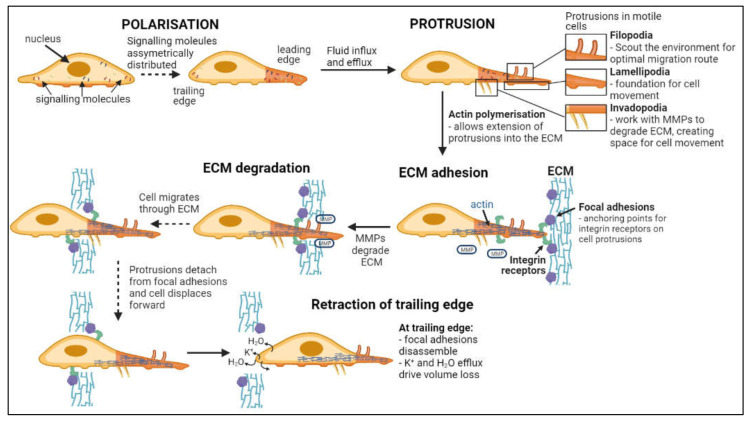
Flow chart of steps in cell invasion. Leading and trailing edges of cells form following asymmetric distribution of signaling molecules [71,72]. Fluid influx and efflux at the leading edge drives the formation of protrusions while actin polymerization facilitates extension into the ECM [73,74,75,76,77]. Focal adhesion proteins within the ECM bind integrin receptors on the cell surface, anchoring the cell to the ECM while metalloproteinases (MMPs) degrade the ECM [78,79]. Continuous detachment and reattachment of these anchorage points allows forward displacement of the cell through degrading ECM [80,81]. Complete disassembly of focal adhesion points and volume loss at the trailing edge terminates cellular invasion [76].

**Figure 3 ijms-22-11909-f003:**
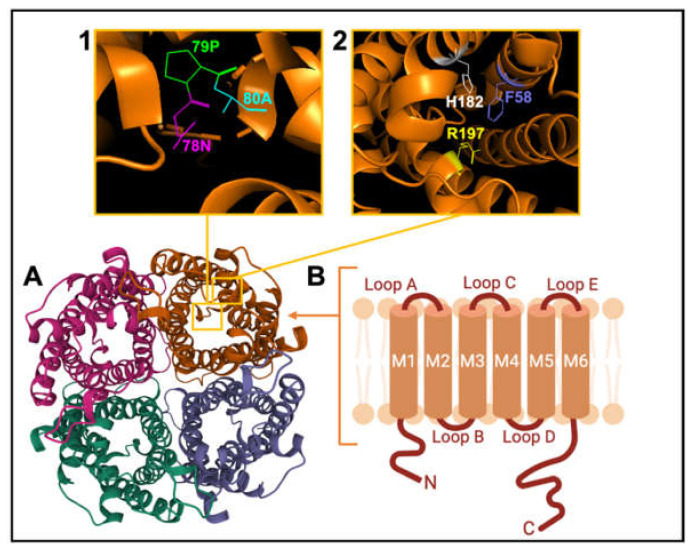
Schematic representation of the AQP channel structure. (**A**). View of the extracellular face of tetrameric AQP1, based on X-ray crystal structure for bovine AQP1 (Protein Data Bank 1J4N). (**B**). Membrane topography of an individual subunit with 6 helical transmembrane domains (M1-M6) connected by loops A to E. (**Inset 1**) View of Pro 79, Asp 80 and Asn 78 residues of the signature NPA motif. (**Inset 2**) View of His 182, Phe 58 and Arg 197 residues of the ar/R constriction selectivity filter.

**Figure 4 ijms-22-11909-f004:**
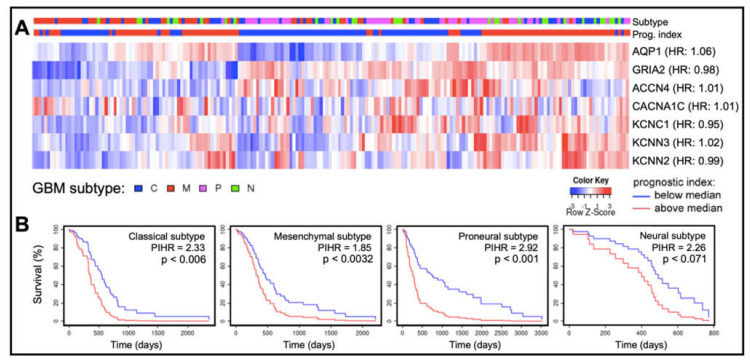
Patterns of expression of ion channels and aquaporin-1 based on transcript levels in human glioblastoma biopsy samples. (**A**) Transcript levels for aquaporin-1 (AQP1), an AMPA glutamate receptor GluR2 (GRIA2), and selected ion channels (acid sensing ion channel-4, ACCN4; Ca_V_ α1C, CACNA1C; K_V_3.1, KCNC1; K_Ca_2.3, KCNN3; and K_Ca_2.2, KCNN2), compiled from the GBM Bio Discovery Portal database (https://gbm-biodp.nci.nih.gov accessed on 26 September 2021), are presented as heat maps for low (blue) and high (red) transcript levels quantified as FPKM (Fragments Per Kilobase of transcript per Million mapped reads, referenced to the ‘Color Key’), and shown as Z-scores for each patient relative to the average transcript level across the sample population. Samples also are identified by glioblastoma subtype (top bar; referenced to the key ‘GBM subtype’). Gene names are annotated with hazard ratios (HR) from Cox analyses (right side of heat map rows). (**B**) Kaplan–Meier survival curves for each glioblastoma subtype for the cluster of genes summarized in (**A**). In classical, mesenchymal and proneural subtypes, multivariate analysis revealed a significant correlation between patient survival time and the prognostic index hazard ratios (PIHR) of collective AQP1, ion channel and AMPA-type glutamate receptor transcript levels (*p* < 0.05).

**Figure 5 ijms-22-11909-f005:**
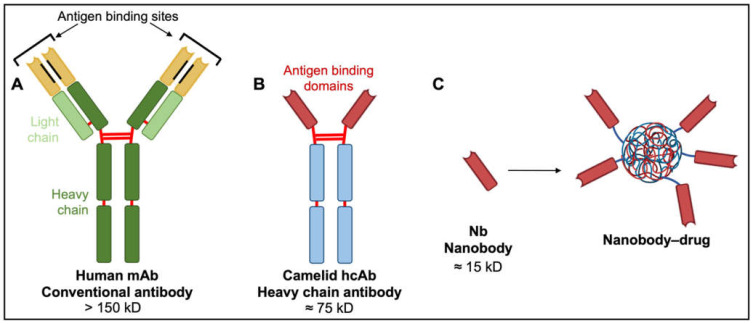
Comparison of human, camelid and synthetic nanobody structures. (**A**) Conventional antibodies have two heavy and two light chains each comprising non-covalently associated variable domains (VH and VL) with unique antigen binding sites. The orientation of these variable domains is mediated by a hydrophobic interface (black lines) and further stabilized by the disulfide-linked CL and CH1 domains (red lines). (**B**) Camelid heavy chain antibodies consist of two heavy chains lacking CH1 domains and containing target-binding modules composed of one variable domain (VHH). (**C**) An isolated VHH region termed a nanobody (Nb) is entirely hydrophilic. Nanoparticles made of liposomes containing anti-neoplastic drugs linked to nanobodies have exciting potential as tumor-targeting drug moieties.

**Table 1 ijms-22-11909-t001:** Molecular markers and genetic hallmarks of the glioblastoma subtypes, defined by Verhaak and colleagues [19] and expanded by Brennan and colleagues [20].

Subtype	Molecular Markers	Genetic Hallmarks
Classical	- Epidermal growth factor receptor (EGFR) upregulation, deletion of cyclin-dependent kinase CDKN2A [19] - Downregulation of pro-apoptotic proteins including Bid, Bak and cleaved caspases 7 and 9, Bid and Bak [20]	- Increased angiogenesis - Regulation of cell cycle progression and cell death lost - Decreased MAP kinase signaling and apoptosis
Mesenchymal	- High frequency of tumor suppressor NF1 inactivation and deletion, mutation of Tumor suppressor PTEN [19] - Upregulation of tumor necrosis family and endothelial markers CD31 and VEGFR-2 [20]	- Constitutive phosphoinositide 3-kinase (PI3K) activity and decreased apoptosis following loss of negative feedback by PTEN - Increased necrotic tissue within tumor
Neural	- Enriched neuronal genes NEFL, GABRA1 and STY1 [19]	- Accelerated infiltration of macrophages and microglia
Proneural	- Amplification of platelet-derived growth factor (PDGF), oncogene *MYC* and cyclin-dependent kinase *CDK4* [20] - Isocitrate dehydrogenases (IDH1 and IDH2) and *transcriptional regulator ATRX* mutated [19,20]	- Increased angiogenesis - Enhanced Wnt/β-catenin signaling - Dysregulation of protein synthesis and the cell cycle - Elevated PI3K and mTORC1 pathway activity

**Table 2 ijms-22-11909-t002:** Reported roles of ion channels in cellular invasion and migration events.

Ion Channel Class	Role(s) in Cellular Motility
Voltage-gated potassium channels	- Modulates cell migration by allowing calcium entry and inducing rapid focal adhesion turnover rates [7] - Regulating expression of RhoC, a key regulator of focal adhesion assembly, and ECM-degrading MMP2 [130]
Voltage-gated chloride channels	- Facilitating the chloride efflux involved in cytoplasm secretion, enabling cell shrinkage [106] - Colocalizing with MMP2 to form a macromolecular complexes that facilitate protein trafficking promotive of ECM degradation [131]
Acid-sensing ion channels	- Facilitating cation conductance to regulate cell volume during tumor cell migration through interstices [132]
Voltage-gated sodium channels	- Altering cytoskeletal elements to adopt cellular morphologies facilitative of migration [133] - Promoting proteolytic degradation of the ECM and cell-cell adhesion [22,134]
Voltage-gated calcium channels	- Facilitating EGFR signaling and ECM stiffening to induce collective cancer cell invasion of the surrounding tumor microenvironment [14] - Activating sheddase ADAM10 function and the downstream Wnt/β-catenin-mediated signaling pathway underlying cellular motility [135]

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
