# Peer review of "Novel Ion Channel Targets and Drug Delivery Tools for Controlling Glioblastoma Cell Invasiveness"

_ijms, 2021, doi:10.3390/ijms222111909_

Round 1

Reviewer 1 Report

This review is a well-detailed description of the mechanisms of glioblastoma motility, and alternative or adjunctive approaches to treatment regimes, particularly in cases of existing drug resistance. The content is timely, convincing, and, I believe, important for the field when considering drug screens or translational studies going forward. 

My main concern with the review is that whilst a lot of the core literature is cited, there appears to be gap in discussing current literature (some of the main areas will be listed below). There are also some areas where bold statements are made without much evidence to support. However, I would consider all of the comments below to be of minor revision.

Line 26: You state that inhibition of AQPs and ion channels collectively would be effective at lower doses, but we don't know that, and isn't really the point your review is trying to make. It might be more useful to propose that co-treatment with these types of inhibitors could offer an alternative approach to overcome limitations with current strategies.

Line 49-51: Cite some of the literature demonstrating effectiveness of inhibiting cell migration in carcinomas in vitro and in vivo models. It might also be interesting to expand on mechanisms that have been demonstrated more robustly in other tumour cell migration studies.

Line 68: "...treatment strategies aimed at eradicating the primary tumors". Ref?

Table 1: There are more genes involved in each of the glioblastoma subtypes that could be included in this table. Potentially some important missing literature here. E.g see Brennan CW, et al. The somatic genomic landscape of glioblastoma. Cell. 2013;155(2):462–77.

Lines 101-144: There's a lot more recent evidence that could be used to enhance this paragraph. A lot of the key literature about EGFR mutations and GBM can be found throughout Liu, X. et al (2019) Journal of Experimental & Clinical Cancer Research (38) 219.

Line 173: "Glioblastoma treatments aimed at mechanisms beyond DNA repair are needed." Sentence isn't needed here, move to the last subparagraph.

Line 209: With the statement from 173, given you've just described a range of sub-effective or problematic existing treatments, it might be nice to emphasize the lack of discovery for novel drug targets over the last 10 years or so, and how this requires us to look at alternative mechanisms.

Figure 2: Could you use black font only? I struggle to read the text, particularly in yellow, and it doesn't add anything. 

Line 348: I'm not entirely sure what you mean by "interchangeable", but if you mean that no other AQP subtypes facilitate GBM cell migration, i'd very much disagree. AQP4 is also implicated in glioblastoma migration. DOIs:

10.1158/0008-5472.CAN-18-2015

10.1158/0008-5472.CAN-19-1185

10.3892/or.2012.1973

10.1038/s41392-019-0103-4

etc.

Line 369: Regulation of AQP4 sub-cellular localization and turnover may also have a role. DOIs:

10.1002/glia.20627

10.1016/j.cell.2020.03.037

10.1074/jbc.M115.646034

Line 465: There isn't a lot of convincing evidence that Bacopasides are actual AQP1 "blockers", or what the PK/PD is like. Oocyte swelling assays have a high rate of artefact. For example, the proposed AQP4 blocker acetazolemide was shown to reduce oocyte swelling, but was later confirmed to have no actual influence on AQP4 inhibition. "Proposed AQP1 inhibitor" would be a better terminology.

Author Response

We thank the reviewer for the careful analyses and excellent comments and suggestions. All concerns have been addressed as suggested in the revised version of the MS.

Line 26: You state that inhibition of AQPs and ion channels collectively would be effective at lower doses, but we don't know that. It might be more useful to propose that co-treatment with these types of inhibitors could offer an alternative approach to overcome limitations with current strategies.

            This section in the Abstract (lines 26-30) has been rewritten as suggested.

Line 49-51: Cite some of the literature demonstrating effectiveness of inhibiting cell migration in carcinomas in vitro and in vivo models. It might also be interesting to expand on mechanisms that have been demonstrated more robustly in other tumour cell migration studies.

            This additional information has been added to the Introduction (lines 68-76).

Line 68: "...treatment strategies aimed at eradicating the primary tumors". Ref?

            This sentence has been edited to simplify the structure, clarify the meaning, and a reference has been added. (line 92).

Table 1: There are more genes involved in each of the glioblastoma subtypes that could be included in this table. Potentially some important missing literature here. E.g see Brennan CW, et al. The somatic genomic landscape of glioblastoma. Cell. 2013;155(2):462–77.

            Additional information from Brennan et al has been added to the completely revised Table 1.

Lines 101-144: There's more recent evidence that could be used to enhance this paragraph. A lot of the key literature about EGFR mutations and GBM can be found throughout Liu, X. et al (2019) Journal of Experimental & Clinical Cancer Research (38) 219.

            Additional information including the work published by Liu et al 2019 and others has been added (lines 166-174).

Line 173: "Glioblastoma treatments aimed at mechanisms beyond DNA repair are needed." Sentence isn't needed here, move to the last subparagraph.

            The sentence has been moved as suggested (now lines 261-262).

Line 209: With the statement from 173, given you've just described a range of sub-effective or problematic existing treatments, it might be nice to emphasize the lack of discovery for novel drug targets over the last 10 years or so, and how this requires us to look at alternative mechanisms.

            This point is now incorporated as new text (lines 259-261).

Figure 2: Could you use black font only? I struggle to read the text, particularly in yellow, and it doesn't add anything.

            Apologies; this change has been made as suggested. (new Figure 2)

Line 348: I'm not entirely sure what you mean by "interchangeable", but if you mean that no other AQP subtypes facilitate GBM cell migration, I'd very much disagree. AQP4 is also implicated in glioblastoma migration.

Line 369: Regulation of AQP4 sub-cellular localization and turnover may also have a role.           

These oversights have been rectified with a new detailed subsection (lines 421-443).

Line 465: There isn't a lot of convincing evidence that Bacopasides are actual AQP1 "blockers", or what the PK/PD is like. Oocyte swelling assays have a high rate of artefact. For example, the proposed AQP4 blocker acetazolemide was shown to reduce oocyte swelling, but was later confirmed to have no actual influence on AQP4 inhibition. "Proposed AQP1 inhibitor" would be a better terminology.

            Evidence for the translational relevance of bacopaside as a demonstrated AQP1 blocker in an in vivo model should have been included. This omission has been corrected. (lines 557-560). 

Reviewer 2 Report

The manuscript by Varricchio et al. provides a review of two seperate aspects regarding the treatment of glioblastoma: Potential drug targets and potential drug delivery methods. Though the authors gave an in-depth review of the both fields, two major concerns remain:

  • With regard to aquaporins and ion channels as potential drug targets, most of the evidence was from in vtro experiments. The author might consider to discuss 1) How relevant are the in vitro models, 2) why it was or still is not possible to test the hypothesis in vivo.
  • With regard to nanobody as drug delivery tool, it is unknown whether it can cross the blood-brain barrier. The biodistribution study cited by the authors didn't include brain. Since blood-brain barrier penetration is essential for a drug delivery tool for the treatment of glioblastoma, this aspect should be included in the evaluation of suitability of nanobody as drug discovery tool.

Author Response

We thank the reviewer for pointing out important gaps in the presentation

With regard to aquaporins and ion channels as potential drug targets, most of the evidence was from in vitro experiments. The author might consider to discuss 1) How relevant are the in vitro models, 2) why it was or still is not possible to test the hypothesis in vivo.

            The important principle of in vivo relevance of AQP blockers and translation from in vitro to in vivo has been expanded in revised MS, as also suggested by Reviewer 1. (lines 557-560)

With regard to nanobody as drug delivery tool, it is unknown whether it can cross the blood-brain barrier. The biodistribution study cited by the authors didn't include brain. Since blood-brain barrier penetration is essential for a drug delivery tool for the treatment of glioblastoma, this aspect should be included in the evaluation of suitability of nanobody as drug discovery tool.

            This concept and new approaches being used to address the barrier challenge have been explained in more detail (lines 678-687)

Reviewer 3 Report

In the present manuscript, the authors examined the glioblastoma multiforme (GBM), a daunting clinical challenge for the capacity of tumor cells to rapidly infiltrate healthy brain parenchyma and resist the common treatment strategies. The difficulties of the current strategies for glioblastoma treatment, also for the presence of different subtypes, were described. The aberrant cell motility underlies the exceptionally invasive nature of these tumors and, until now, its management remains scarce. The potential for combined inhibition of AQPs and ion channels to counteract cell motility was described. Finally, the authors suggest the use of nanobodies to selectively target and deliver drugs to the tumor cells, overcoming off-target effects in the brain environment.

The description is accurate, accompanied by useful tables and figures. Overall, a good amount of literature was examined and described clearly. The review is well written, attractively presented, and updated.

Specific comments:

  1. ECM should be abbreviated when first cited.
  2. AQP11 also has permeability to H2O2. Bestetti et al., 2020 should be cited (Human aquaporin-11 guarantees efficient transport of H2O2 across the endoplasmic reticulum membrane. Bestetti S, Galli M, Sorrentino I, Pinton P, Rimessi A, Sitia R, Medraño-Fernandez I. Redox Biol. 2020 Jan; 28:101326. doi: 10.1016/j.redox.2019.101326. Epub 2019 Sep 12. ).

Author Response

We appreciate the reviewer's helpful suggestions and corrections.

ECM should be abbreviated when first cited.

            This definition has been added at the first mention in the text (lines 277-278).

AQP11 also has permeability to H2O2.

            This property and the appropriate references have been added (lines 351-352).

Round 2

Reviewer 2 Report

Major concerns have been addressed in the revised manuscript.